# Partograph utilization and associated factors among obstetric care givers in governmental health institutions of Jigjiga and Degehabur towns, Somali region, Ethiopia: A cross-sectional study

Liyew Mekonen Ayehubizu[1]*, Abebe Tadesse Tibebeu[2], Metsihet Tariku Fetene[3], Semehal Haile Yohannes[4], Zemenu Shiferaw Yadita[5]

1 Department of Public Health, College of Medicine and Health Sciences, Jigjiga University, Jijiga, Ethiopia,
2 Department of Midwifery, College of Medicine and Health Sciences, Jigjiga University, Jijiga, Ethiopia,
3 Department of Nursing, College of Medicine and Health Sciences, Jigjiga University, Jijiga, Ethiopia,
4 Department of Neonatal Nursing, College of Medicine and Health Sciences, Jigjiga University, Jijiga, Ethiopia, 5 Department of Reproductive Health and Population Studies, College of Medicine and Health Science, School of Public Health, Bahir Dar University, Bahir Dar, Ethiopia

* liy900@gmail.com

**Data Availability Statement:** All relevant data are within the paper and its Supporting information files.

## Abstract

### Background

Partograph is a simple, inexpensive & economical tool that provides a continuous graphical overview of labour and prevents prolonged and obstructed labor. The purpose of the study is to assess partograph utilization and associated factors among obstetric care givers in governmental health institutions of Jigjiga and Degehabur Towns, Somali Region, Ethiopia.

### Methods

An institution based cross-sectional quantitative study was carried out among obstetric care givers who were working in governmental health institutions. Systematic random sampling with proportional to size allocation was used to recruit a total of 235 study participants. Self-administered questionnaire was used to collect data in this study. Three data collectors and one supervisor were recruited and trained to facilitate the data collection activities. Data were entered into Epi data software and exported into SPSS (23.0) for analysis. Descriptive statistics, bivariate and multivariate logistic regression were computed to determine proportions and significant association with partograph utilization among obstetric care givers.

### Result

Less than half of obstetric care givers, 41% (95%CI: 34.5–46.9) had good partograph utilization to monitor progress of labor. Being female [AOR = 2.36, 95%CI:(1.03–5.44)], availability of partograph [AOR = 4.633, 95%CI: (1.698–12.640)], having good knowledge [AOR = 6.90, 95%CI:(2.62–18.18)], receiving on job training [AOR = 15.46, 95%CI:(6.95–34.42)] and

**Funding:** The author(s) received no specific funding for this work.

**Competing interests:** The authors have declared that no competing interests exist.

**Abbreviations:** ANC, Ante Natal Care; AOR, Adjusted Odds Ratio; APGAR, Activity, Pulse, Grimace, Appearance &Respiration; CI, Confidence Interval; COR, Crude Odds Ratio; CSA, Central Statistical Agency; EDHS, Ethiopian Demographic and Health Survey; FHR, Fetal Heart Rate; ICM, International Council of Midwives; IMB, Integrated Model of Behavior; MCH, Maternal and Child Health; MM, Maternal Mortality; MMR, Maternal Mortality Rate; SD, Standard Deviation; SMI, Safe Motherhood Initiatives; SPSS, Statistical Package for Social Sciences; UN, United Nation; UNFPA, United Nations Population Fund Affairs; UNICEF, United Nations Children Fund; USA, United State of America; WHO, World Health Organization.

positive attitude towards partograph [AOR = 2.99, 95%CI:(1.25–7.14)] were significantly associated with partograph utilization.

## Conclusion

Partograph utilization in this study was low. Especial emphasizes and interventions should be given to periodic on job training that improve knowledge and attitude of obstetric care givers to increase partograph utilization.

## Background

Women are suffering many health problems during their pregnancy as well as at the time of child birth/delivery; globally, there were an estimated number of 15% expected births that developed life-threatening complications during pregnancy, delivery or the postpartum period [1–4].

In 2017, about 295 000 women died during and following pregnancy and childbirth. Sub-Saharan Africa alone accounted for roughly two-thirds (196 000) of maternal deaths, while Southern Asia accounted for nearly one-fifth (58 000); and both accounted for approximately 86% (254 000) of the estimated global maternal deaths [5]. According to the 2015 Ethiopia Demographic and Health Survey, maternal mortality ratio was 353 per 100,000 live births in Ethiopia [6–9]. Though it was slightly increased to 412 in 2016 [10]. Maternal mortality was still high in 2019 accounted to 412 per 100,000 live births [11].

The major causes of maternal and neonatal death in Ethiopia are obstructed labour and prolonged labour which was account for (13%) of maternal deaths [9, 12]. Obstructed and prolonged labour contribute to series of complication in mothers and newborn infants if it is not managed early [13]. These complications can be prevented by proper utilization of partograph during the progress of labour and providing skill delivery services [14].

The partograph is a graphical presentation of the progress of labour, and of fetal and maternal condition during labour [2, 12, 15–17]. It consists of four sections: the maternal information, the fetal conditions record, the labor progress record, and the maternal conditions record [17–19]. The fetal condition record may track fetal heart rate, amniotic liquor, and molding of the fetal skull. The labor progress record tracks cervical dilatation and descent of the fetus' head over time, comparing it to a pre-printed "alert" and "action" lines [2, 9, 15, 18]. The maternal conditions record often captures contractions, blood pressure, pulse, urine output, temperature, and drugs administered including drugs to help the uterus contract [12, 17–20].

Universal utilization of the partograph during labour is recommended by world health organization since routine use of partograph is helpful to make better decisions for the diagnosis and management of prolonged and obstructed labour [17]. It is still not broadly used in the developing countries especially in Africa include Ethiopia. Studies done in Baghdad, a metropolitan area in Ghana and Lusaka: in Zambia, showed that58%, 54% and 87.5% of the participants used partograph to monitor progress of labour respectively [21–23]. In Ethiopia, there is no consistent use of the partograph during labour; Studies done in Asella referral and teaching hospital, Sidama zone, Bale zone, East Gojjam zone, Addis Ababa city administration; showed that 26%, 50.7%, 70.2%, 53.85% and 69% of the participants used partograph to monitor progress of labour respectively [1, 9, 14–16]. This inconsistent use of partograph due to different factors such as participant's sex, age, year of clinical service, health profession, knowledge

about partograph, attitude towards partograph, absence of training on partograph, work place and work over load [1, 2, 12, 14, 15, 17, 21, 24].

Assessing obstetric care giver's practical utilization of partograph and its determinants has great value to design appropriate intervention strategies to provide quality maternity care. Therefore, the purpose of this study was to assess utilization of partograph and associated factors among obstetric care givers working in governmental health institutions of Jigjiga and Degehabur towns, Somali region, Ethiopia.

## Methods

### Study setting and period

The study was conducted in public health institutions Jigjiga and Degehabur towns from May18/2020 –July 8/2020. Jigjiga town, a capital city of Somali Region, is located 635kms east of Addis Ababa. It has 20 kebeles. The town administration has two hospitals, two health centres and fourteen health posts. In the town there are 1,208 clinical health professional workers (212 midwives, 11 HO, 137 medical doctors and 6 Gynecologist and obstetrician) that work in obstetric care at governmental health institutions. It has 366 total numbers of obstetric care givers.

Degehabur is a town in the Somali Region of Ethiopia, is located 800 &165 kms north of Addis Ababa and Jigjiga town respectively. It is the administrative centre of Degehabur) woreda. The town administration has 1 hospital and 1 health centre of public health institutions. It has also 258 clinical health professional workers. Among those 51 midwives, 9 Health officer, 10 medical doctors, 1 Emergency Surgery and 1 obstetrician that work in obstetric care in the town of public health institutions. It has 72 total numbers of obstetric care givers. Total number of obstetric care givers in two towns is 438 that work public health institutions.

### Study design

An institution based cross-sectional study design was conducted to assess utilization of partograph and associated factors.

### Study population

All obstetric care providers (midwives, nurses, general practitioners, and health offers working in governmental health institutions) in Jigjiga and Degehabur Towns, Somali Region, Ethiopia included in the study.

### Sample size determination

The sample size was estimated using single proportion formula by assuming 5% marginal error and 95% confidence interval ($\sigma = 0.05$) and study conducted in central zone, Tigray in which the proportion of utilization of partograph among obstetric care givers were 73.3% [2]. By adding 10% for non-respondents the final sample size was taken as 196. Moreover, the double population proportion formula was used to determine the sample size for factors associated with partograph utilization to get maximum possible sample size. It was calculated for some of the associated factors obtained from different literatures using Epi info statistical software version 7 with the following assumptions: confidence level = 95%, power = 80%, the ratio of unexposed to exposed almost equivalent to 2 and partograph utilization among those do not get additional training is 56.5% with AOR 2.4 [14]. Therefore, the final estimated sample size is 235 after adding 10% non response rate which was higher than the sample size calculated using single population proportion formula.

## Sampling procedure

From all health institutions found in Jigjiga and Degehabur towns, six health institutions (Jigjiga university Sheik Hassen Yabare Referal Hospital, Kharamara Hospital, Ablele health center, Ayerdega health center, Degehabour General Hospital, and Degehabour health center) were selected using a simple random sampling technique.

The total number of health professionals found in selected hospitals and health centers in Jigjiga town were 234 and 50 respectively. Additionally, 51 health professionals from the Degehabour General hospital and 21 from Degehabour health center were included. The required number of study subjects was calculated via proportional allocation, with 187 from Jigjiga and 48 from Degehabour. Study subjects were chosen using a simple random sampling procedure (lottery method) after preparing sampling frame.

## Data collection instrument and techniques

Semi—structured questionnaire which was developed from different literatures was used [2, 16, 17, 21, 22, 24]. It was prepared in English version. Three Diploma in Midwifery data collectors and one BSc in midwifery supervisor were hired and trained for data collection.

## Data processing and analysis

Data was entered, checked, and analyzed using SPSS version 23.0. Descriptive statistics was employed to calculate frequencies, median and percentage. Attitude was measured as follows: there were six (6) attitude determining Likert scale type questions prepared to assess respondents' attitude towards partograph. The responses of "strongly agree was scored 5, agree was scored 4, uncertain was scored 3, disagree was scored 2 & strongly disagree was scored 1. The scoring was reversed for negative statements. The scores of the items were summed-up and the total was divided by the number of the items, giving a mean score. These score was converted into a percent score and mean were computed. The attitude was considered "positive" if the scored 60% or more and "negative" if less than 60%.

Bivariable logistic regression analysis was made using OR and 95% CI to assess the association of independent variable with the outcome separately. Based on Bivariate analysis variables that showed significant association at ($p < 0.2$) were entered to multivariable analysis to select Predictor variables of factors affecting partograph utilization. Finally, variables that showed significant association at ($p < 0.05$) were identified as independent predictors of partograph utilization.

## Ethical considerations

The Ethical approval was obtained from Institutional Review Board of College of Health Sciences, Jigjiga University on May 14/2020 with reference number RERC/016/2020. Communications with relevant bodies was made through a formal letter obtained from regional health. The objective and importance of the study was explained to the study participants. Data was collected after full informed written consent was obtained from participants. Confidentiality of the information and privacy is maintained.

# Results

## Socio-demographic characteristics of study participants

Two hundred twenty eight respondents completed all the questionnaires correctly making a response rate of 97.02%. About 70.6% of obstetric care givers were females. The median ages of the respondents were 26.00 (±3 IQR) years and more than half 56.1% of them were within

**Table 1. Socio-demographic characteristics of obstetric care givers in governmental health institutions of Jigjiga and Degehabur Towns, Somali region, Ethiopia, 2020 (n = 228).**

| Variables | Frequency | Percent |
|---|---|---|
| **Sex** | | |
| Male | 67 | 29.4 |
| Female | 161 | 70.6 |
| **Age group** | | |
| 20–24 | 54 | 23.7 |
| 25–29 | 128 | 56.1 |
| 30 and older | 46 | 20.2 |
| **Professional qualification** | | |
| Nurse | 28 | 12.2 |
| Midwife | 152 | 66.7 |
| HO (health officer) | 17 | 7.5 |
| General practitioner (MD) | 31 | 13.6 |
| **Year of service** | | |
| ≤4 | 156 | 68.4 |
| 5–10 | 55 | 24.1 |
| ≥11 | 17 | 7.5 |
| **Place of work** | | |
| Health center | 66 | 28.9 |
| Hospital | 162 | 71.1 |

the age group 25–29 years. Regarding their profession, 66.7% were midwives followed by medical practitioner (MD) 13.6%. Regarding to participant's working place, 71.1% of them were working at hospitals and the rest 25.0% were working in health centers. More than two third of the respondents 68.4% had ≤4 clinical service years and median year of clinical service of obstetric care givers were 3.00 (±3 IQR) years (Table 1).

## Knowledge of obstetric care givers about partograph

All participants had heard about partograph while 52.6% of them know the exact definition of partograph. Regarding to starting time of plot on partograph, 83.3% of the participants correctly identified the exact time when to start plotting partograph. However, 61.4% of the respondents did not know the component of partograph and also 71.5%, 68%, 76.8% of the respondents did not know the definition of alert line, action line and satisfactory of labour progress respectively. Pertaining to use of partograph, most of the participant 78.9% mentioned that partograph should be used for all women in active first stage of labor. Majority (63.6%) of respondents said they had not received any photographs on the job training (Table 2).

Regarding respondents' knowledge about partograph, only less than one third of them 28.5% had good knowledge (Fig 1).

## Attitude of obstetric care givers towards partograph utilization

Less than half of them (46%) strongly agreed that the photograph is beneficial. More than one third of participants, 45.2% of them were agreed that partograph is favorable as it alerts obstetric care givers of any deviation from normal and 43.9% of the respondents agreed that health care givers are able to identify problems and recognize complications early. Less than one

**Table 2. Knowledge related responses of obstetric care givers about partograph in governmental health institutions of Jigjiga and Degehabur Towns, Somali region, Ethiopia, 2020 (n = 228).**

| Variables | Frequency | Percent |
|---|---|---|
| **Heard about partograph** | | |
| Yes | 228 | 100.0 |
| **Definition of partograph** | | |
| A tool to be used only in active phase of labor | 120 | 52.6 |
| A graphic method of recording first stage of labor | 71 | 31.2 |
| A salient feature of recording the whole process of labor | 37 | 6.2 |
| **Component of the partograph** | | |
| List all | 88 | 38.6 |
| Not list all | 140 | 61.4 |
| **Knowledge about the start time of plotting partograph** | | |
| At 4cm cervical dilatation | 190 | 83.3 |
| At 3cm cervical dilatation | 18 | 7.9 |
| When labor is diagnosed | 20 | 8.8 |
| **Definition of alert line** | | |
| Define correctly | 65 | 28.5 |
| Not define correctly | 163 | 71.5 |
| **Definition of action line** | | |
| Define correctly | 73 | 32 |
| Not define correctly | 155 | 68 |
| **Definition of satisfactory of labor progress** | | |
| Define correctly | 53 | 23.2 |
| Not define correctly | 175 | 76.8 |
| **Detected any complication using a partograph** | | |
| Yes | 196 | 86 |
| No | 32 | 14 |
| **List one complication if you detect using partograph** | | |
| List | 180 | 91.8 |
| Not list | 16 | 8.2 |
| **Type of client that needs partograph use** | | |
| Primgravid | 17 | 7.5 |
| Multiparious | 16 | 7 |
| Eclamptic patients | 15 | 6.6 |
| All women in active labour | 180 | 78.9 |
| **On job training** | | |
| Yes | 83 | 36.4 |
| No | 145 | 63.6 |

third (26.3%) of them agreed that skilled birth attendant must use a partograph for every labouring mother. Near to thirty one percent (30.7%) of them were uncertain and 28.5% were strongly disagreed that using partograph enables health care givers perform essential basic interventions. And also 35.1% of them were strongly disagreed that using partograph misleads management as the progress of labour and the partograph alert line are not aligned in most pregnant women (Table 3).

Regarding respondents' attitude towards partograph utilization, only less than one third of them (18%) had positive attitude (Fig 2).

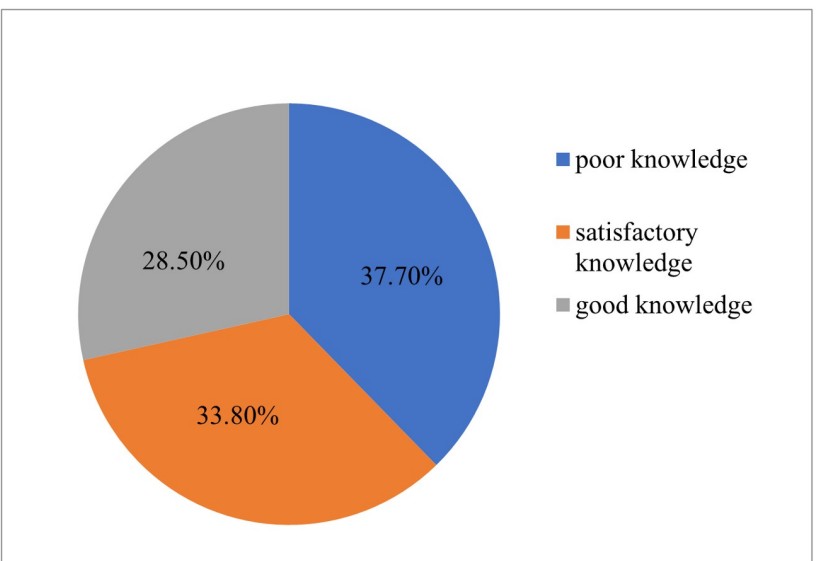

**Fig 1. Knowledge of obstetric care givers about pantograph in governmental health institutions of Jigjiga and Degehabur Towns, Somali region, Ethiopia, 2020 (n = 228).**

### Reasons for not using the partograph

As indicated in Fig 3, less than half (42.5%) of respondents said that lack of orientation was the reason for not using the partograph when monitoring women in labor.

### Partograph utilization

In this study, majority (74.1%) of the respondents had reported the availability of partograph in the facility. More than half (64%) of the participants were partially recorded parameters of partograph on the tool. Similarly, (26.8%) of the participants were conducted deliveries from 7 to 9 in the institution per shift day. Most (92.1%) of the participants reported that they hadn't faced any problem to utilize partograph. Majority (68.4%) of participants had work load in their facility. More than three-fourths (76.3%) gave advice for the obstetric care givers to encourage use partograph during labor (Table 4).

Overall from 228 participants, 93(41%) of the participants were had good partograph utilization (Fig 4).

### Factors associated with partograph utilization

Bivariate analysis was performed to select candidate variables for multivariable analysis. Accordingly, sex of participant (being female) [COR = 2.15, 95%CI: (1.16–3.98)], working in Hospital [COR = 1.71, 95%CI: (0.94–3.14)], having good knowledge [COR = 2.42, 95%CI: (1.24–4.70)], availability of partograph [COR = 6.45, 95%CI: (2.89–14.41)], taking on job training [COR = 12.08, 95%CI: (6.34–22.99)] and positive attitude towards partograph [COR = 2.73, 95%CI: (1.37–5.48)] were among factors showed that candidate variables for multivariate analysis at p value</ = 0.2.

Multivariate analysis was performed for the purpose of controlling the multiple confounding effects of variables. Accordingly, sex of participant (being female) [AOR = 2.36, 95%CI: (1.03–5.44)], availability of partograph [AOR = 4.63, 95%CI: (1.70–12.64)], having good knowledge [AOR = 6.90, 95%CI:(2.62–18.18)], receiving on job training [AOR = 15.46, 95%

**Table 3. Attitude related responses of obstetric care givers towards partograph utilization governmental health institutions of Jigjiga and Degehabur Towns, Somali region, Ethiopia, 2020 (n = 228).**

| Variables | Frequency | Percent |
|---|---|---|
| **Partograph is beneficial** | | |
| Strongly agree | 105 | 46 |
| Agree | 95 | 41.7 |
| Uncertain | 13 | 5.7 |
| Strongly disagree | 15 | 6.6 |
| **Partograph is very favorable as it alerts health workers any deviation from normal** | | |
| Strongly agree | 50 | 22 |
| Agree | 113 | 49.6 |
| Uncertain | 45 | 19.7 |
| Disagree | 14 | 6.1 |
| Strongly disagree | 6 | 2.6 |
| **By using partograph, health care providers are able to identify problems and recognize complications early** | | |
| Strongly agree | 31 | 13.6 |
| Agree | 100 | 43.9 |
| Uncertain | 70 | 30.7 |
| Disagree | 14 | 6.1 |
| Strongly disagree | 13 | 5.7 |
| **Skilled birth attendant must use partograph on every laboring mother** | | |
| Strongly agree | 26 | 11.4 |
| Agree | 60 | 26.3 |
| Uncertain | 90 | 39.5 |
| Disagree | 39 | 17.1 |
| Strongly disagree | 13 | 5.7 |
| **Using partograph enables health care givers perform essential basic interventions and make referrals to appropriate levels of care when necessary** | | |
| Strongly agree | 10 | 4.4 |
| Agree | 23 | 10.1 |
| Uncertain | 70 | 30.7 |
| Disagree | 60 | 26.3 |
| Strongly disagree | 65 | 28.5 |
| **Using partograph misleads management as the progress of labour and the partograph alert line are not aligned in most pregnant woman** | | |
| Strongly agree | 16 | 7 |
| Agree | 22 | 9.7 |
| Uncertain | 50 | 21.9 |
| Disagree | 60 | 26.3 |
| Strongly disagree | 80 | 35.1 |

CI:(6.95–34.42)] and positive attitude towards partograph [AOR = 2.99, 95%CI:(1.25–7.14)] were factors showing significant association with partograph utilization at p-value</ = 0.05 (Table 5).

## Discussion

The proportion of partograph utilization was 41% (95%CI: 34.6–46.9). This finding is inconsistent with study conducted in Baghdad (58%), Metropolitan area in Ghana (54%), Bale zone

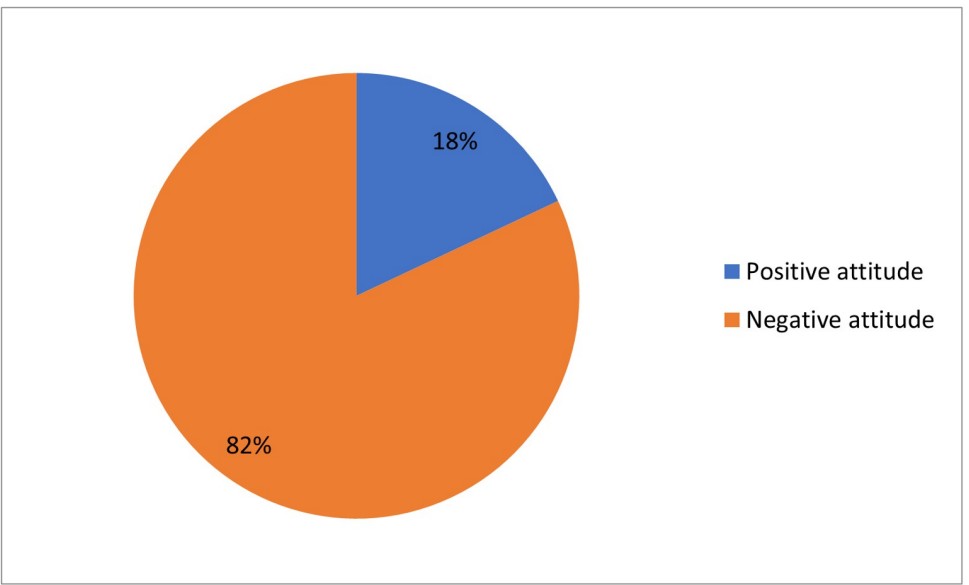

**Fig 2. Attitude of obstetric care givers towards partograph utilization governmental health institutions of Jigjiga and Degehabur Towns, Somali region, Ethiopia, 2020 (n = 228).**

(70.2%), North Showa Zone (81.1%), West Showa Zone (84.6%) and Central Tigray Zone (73.3%) [2, 12, 16, 17, 21, 22]. The lack of well-designed and integrated programs, such as mentorship and supportive supervision, could be the cause of this difference. Other factors that may contribute to lower partograph utilization include a lack of expertise, a lack of understanding, insufficient partograph training, and a negative attitude among study participants.

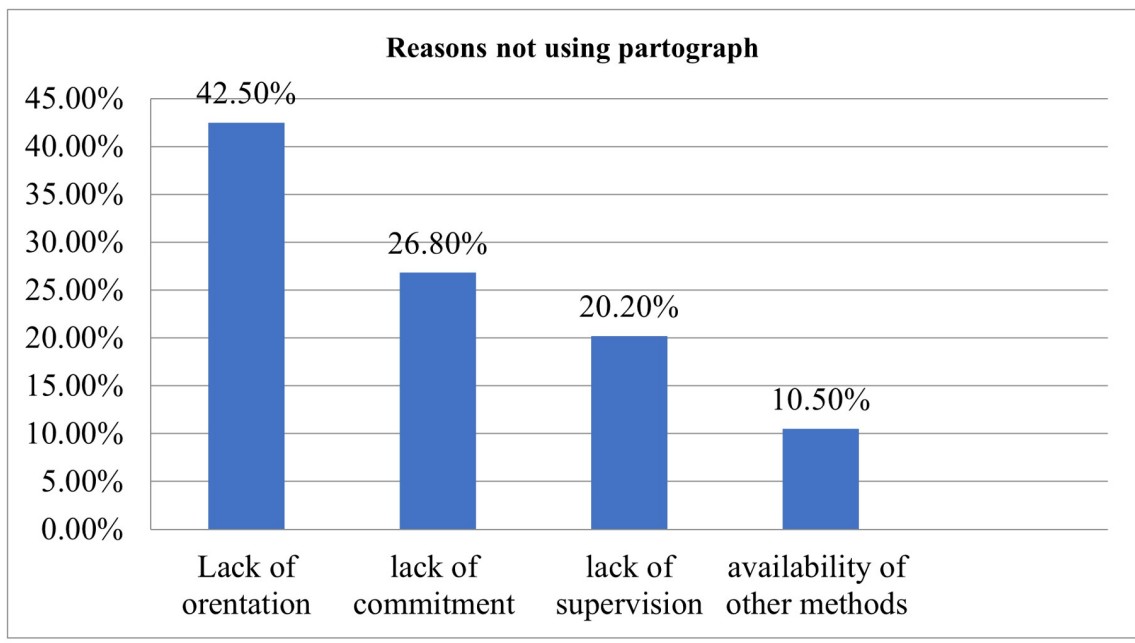

**Fig 3. Reasons for not using the partograph by obstetric care givers in governmental health institutions of Jigjiga and Degehabur Towns, Somali region, Ethiopia, 2020 (n = 228).**

**Table 4. Partograph utilization related responses of obstetric care givers in governmental health institutions of Jigjiga and Degehabur Towns, Somali region, Ethiopia, 2020 (n = 228).**

| Variables | Frequency | Percent |
|---|---|---|
| **Availability of partograph** | | |
| Yes | 169 | 74.1 |
| No | 59 | 25.9 |
| **Partograph use currently** | | |
| Yes | 199 | 87.3 |
| No | 29 | 12.7 |
| **Frequency of partograph utilization** | | |
| Routinely | 71 | 31.1 |
| Sometimes | 128 | 56.1 |
| Occasionally | 29 | 12.7 |
| **Components of partograph plotted** | | |
| Complete | 80 | 35.1 |
| Incomplete | 148 | 64.9 |
| **Number of deliveries conducted per shift day** | | |
| $\leq 3$ | 60 | 26.3 |
| 4 to 6 | 55 | 24.1 |
| 7 to 9 | 61 | 26.8 |
| 10 and above | 52 | 22.8 |
| **Reasons for not utilizing partograph during labor** | | |
| Lack of orientation | 97 | 42.5 |
| Availability of other methods | 24 | 10.5 |
| Lack of commitment | 61 | 28.8 |
| Lack of supervisions | 46 | 20.2 |
| **Facing any problem to utilize the partograph** | | |
| Yes | 18 | 7.9 |
| No | 210 | 92.1 |
| **Work load** | | |
| Yes | 185 | 81.1 |
| No | 43 | 18.9 |
| **Suggestion in order to encourage partograph utilization** | | |
| Give suggestion | 174 | 76.3 |
| Not give suggestion | 54 | 23.7 |

This finding is higher than the study conducted in Asella, Teaching and Referral Hospital, Ethiopia [1]. These discrepancies may be due to the sample size, different study area and time of study.

Findings of this study also indicated that positively significant association between sex and partograph utilization. Female obstetric care givers were more than two times more likely to good partograph utilization than males. This finding is consistent with a previous study done in Bale zone, Ethiopia [16]. It is understandable that the majority of birth attendants assigned to the labor ward on the study area are female midwives who have had the opportunity to participate in partograph usage trainings, which have increased their knowledge and abilities.

Availability of the partograph in health facility is also associated with its utilization. Those health care givers who had partograph tools in their institution were more than four times more likely to use partograph than had not partograph in their facility. This finding is consistent with a previous study done in Asella, Ethiopia and West Showa Oromia, Ethiopia [1, 12].

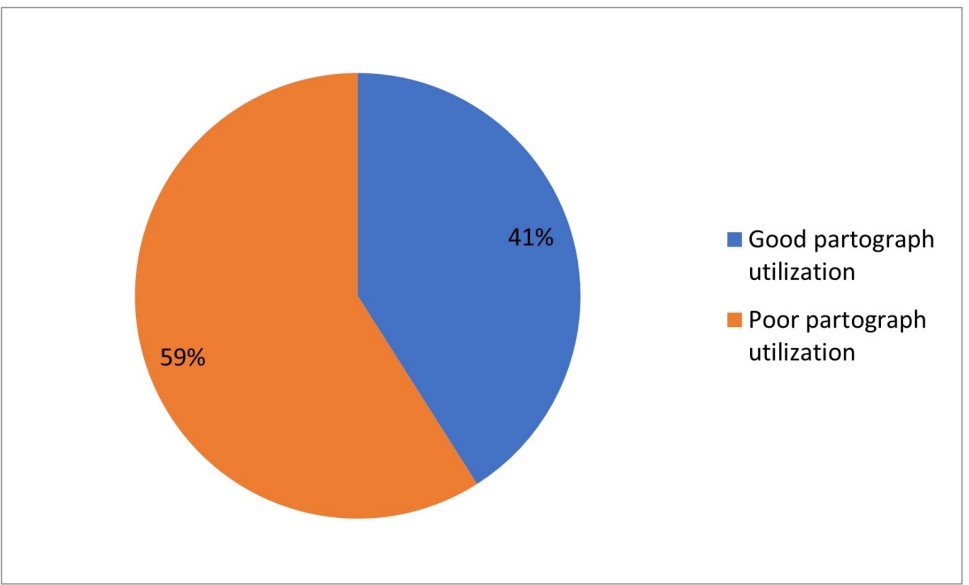

**Fig 4. Utilization of partograph by obstetric care givers in governmental health institutions of Jigjiga and Degehabur Towns, Somali region, Ethiopia, 2020 (n = 228).**

**Table 5. Bivariate and multivariable analysis of factors associated with partograph utilization of obstetric care givers in governmental health institutions of Jigjiga and Degehabur Towns, Somali region, Ethiopia, 2020 (n = 228).**

| Variables | Partograph utilization | | COR | AOR (95%CI) | p-value |
|---|---|---|---|---|---|
| | Yes | No | | | |
| | No (%) | No (%) | | | |
| **Sex** | | | | | |
| Female | 70(47.9) | 76(52.1) | 2.37 (1.32–4.22) | 2.37(1.03–5.41)* | 0.043 |
| Male | 23(28.0) | 59(72.0) | 1 | 1 | |
| **Work place** | | | | | |
| Hospital | 74(45.1) | 90(54.9) | 1.71(0.94–3.14) | 1.35(0.58–3.15) | 0.492 |
| Health center | 21(31.8) | 45(68.2) | 1 | 1 | |
| **Availability of partograph** | | | | | |
| Yes | 85(50.3) | 84(49.7) | 6.45 (2.89–14.41) | 4.63(1.70–12.64)** | 0.003 |
| No | 8(13.6) | 51(86.4) | 1 | 1 | |
| **Knowledge about partograph** | | | | | |
| Good | 35(53.8) | 30(46.2) | 2.42(1.24–4.70) | 6.90 (2.62–18.18)*** | 0.000 |
| Satisfactory | 30(39.0) | 47(61.0) | 1.32(0.70–2.52) | 2.67(1.07–6.71)* | 0.036 |
| Poor | 28(32.6) | 58(67.4) | 1 | 1 | |
| **On job training** | | | | | |
| Yes | 63(75.9) | 20(24.1) | 12.08(6.34–22.99) | 15.46(6.95–34.42)*** | 0.000 |
| No | 30(20.7) | 115(79.3) | 1 | 1 | |
| **Attitude towards partograph utilization** | | | | | |
| Positive | 25(61.0) | 16(39.0) | 2.73(1.37–5.48) | 2.99(1.25–7.14)* | 0.014 |
| Negative | 68(36.4) | 119(63.6) | 1 | 1 | |

Significant at

*p-value<0.05,

**p-value <0.01 and

***p-value <0.001.

This implies that the availability of partograph by itself is essential to motivate obstetrics giver for use of this tools starting from active first stage of labour.

Findings of this study also indicated that significant association between knowledge and partograph utilization. Those obstetric care givers who had good knowledge about partograph were more than six times more likely to good partograph utilization than their counterparts. This finding is consistent with a previous study conducted in Enugu Metropolis, North Showa Zone, Ethiopia, West Showa Zone, Oromia, Ethiopia and Bale zone [12, 16, 17, 24]. The possible reason might be Obstetrics care givers who have knowledge on purpose of partograph may use it more frequently to prevent prolonged labour and related complications.

This research finding showed that on-job training on partograph had a significant association with partograph utilization. Those obstetric care givers who received on job training on partograph were more than fifteen times more likely to utilize partograph than who had not received on- job training. This finding is supported by the study conducted in West Showa zone Oromia Ethiopia, Addis Ababa city administration, Bale zone, East Gojjam zone, central Tigray zone, Easter Ethiopia, North Showa zone and Sidama zone [2, 9, 12, 14–17, 20]. On job trainings enhance knowledge; and improve attitude and skills which in turn increase partograph utilization.

Those obstetric care givers who had positive attitude towards partograph were three times more likely to utilize partograph than to their counterparts. This finding is consistent with a previous study done in North Showa Zone, Ethiopia [17]. The possible reason might be those who had good attitude toward the use of partograph might be committed to improve their skill in use.

## Conclusion

This study revealed that the proportion of partograph utilization by the study participants was low according to Modified WHO partograph and previous studies. Being female obstetrics care giver, availability of partograph, having good knowledge about partograph, having on job training and having positive attitude of participants towards partograph were significantly associated with partograph utilization. Therefore, in this study, we suggest to Jigjiga and Degehabur town's health offices for sustainable and accessible supply of partograph sheets to all institutions to maintain consistency of its utilization Providing periodic on-job training by stakeholders on partograph is also mandatory to improve knowledge and attitude of obstetric care givers towards partograph utilization. Partograph should be also used by obstetric care givers for all laboring women and taken seriously by the care givers and it should be considered as a tool for diagnosing problems, perform essential basic interventions and make referral to appropriate levels of care when necessary, during the progress of labour. Moreover, Further researchers have to be done using qualitative research methodology and on private health institutions to get a comprehensive picture.

## Supporting information

**S1 File. Partograph SPSS file.**
(RAR)

## Acknowledgments

We appreciate Jigjiga University, College of medicine and Health Sciences, School of Graduate Studies, department of Maternity Nursing for giving us this opportunity. We would like to thank Karamara hospital administration office, Jigjiga University Shuik Hassan Yabare

Referral Hospital, Jigjiga City health office and Degehabur town health office staffs; for giving base line information for this study. Finally, we would like to thank data collectors for the contribution of data collection and we would like to thank participates that voluntary involve in this study. We would also like to extend our appreciation to those who has helped me a lot in giving additional advice.

## Author Contributions

**Conceptualization:** Liyew Mekonen Ayehubizu, Abebe Tadesse Tibebeu, Metsihet Tariku Fetene, Zemenu Shiferaw Yadita.

**Data curation:** Liyew Mekonen Ayehubizu, Abebe Tadesse Tibebeu, Metsihet Tariku Fetene, Semehal Haile Yohannes.

**Formal analysis:** Liyew Mekonen Ayehubizu, Abebe Tadesse Tibebeu, Metsihet Tariku Fetene, Semehal Haile Yohannes, Zemenu Shiferaw Yadita.

**Funding acquisition:** Abebe Tadesse Tibebeu.

**Investigation:** Liyew Mekonen Ayehubizu, Abebe Tadesse Tibebeu, Metsihet Tariku Fetene.

**Methodology:** Liyew Mekonen Ayehubizu, Abebe Tadesse Tibebeu, Metsihet Tariku Fetene, Semehal Haile Yohannes.

**Project administration:** Liyew Mekonen Ayehubizu, Abebe Tadesse Tibebeu.

**Resources:** Abebe Tadesse Tibebeu.

**Software:** Liyew Mekonen Ayehubizu, Abebe Tadesse Tibebeu, Metsihet Tariku Fetene, Semehal Haile Yohannes.

**Supervision:** Abebe Tadesse Tibebeu.

**Validation:** Abebe Tadesse Tibebeu.

**Visualization:** Abebe Tadesse Tibebeu, Zemenu Shiferaw Yadita.

**Writing – original draft:** Abebe Tadesse Tibebeu.

**Writing – review & editing:** Liyew Mekonen Ayehubizu, Abebe Tadesse Tibebeu, Metsihet Tariku Fetene, Semehal Haile Yohannes, Zemenu Shiferaw Yadita.

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
