## [Decision Letter · Decision Letter 0]

25 Aug 2021

PONE-D-21-20057

Partograph utilization and associated factors among obstetric care givers in governmental health institutions of Jigjiga and Degehabur towns, Somali region, Ethiopia: a cross-sectional study

PLOS ONE

Dear Dr. Ayehubizu,

Thank you for submitting your manuscript to PLOS ONE. After careful consideration, we feel that it has merit but does not fully meet PLOS ONE’s publication criteria as it currently stands. Therefore, we invite you to submit a revised version of the manuscript that addresses the points raised during the review process.

We look forward to receiving your revised manuscript.

Kind regards,

Nülüfer Erbil, Ph.D, Prof.

Academic Editor

PLOS ONE

Journal Requirements:

2. Please include additional information regarding the survey or questionnaire used in the study and ensure that you have provided sufficient details that others could replicate the analyses. For instance, if you developed a questionnaire as part of this study and it is not under a copyright more restrictive than CC-BY, please include a copy, in both the original language and English, as Supporting Information

4. Please include your tables as part of your main manuscript and remove the individual files. Please note that supplementary tables (should remain/ be uploaded) as separate "supporting information" files

5. Thank you for submitting the above manuscript to PLOS ONE. During our internal evaluation of the manuscript, we found significant text overlap between your submission and the following previously published works, some of which you are an author.

-https://bmcresnotes.biomedcentral.com/articles/10.1186/s13104-018-3814-7

-https://www.hindawi.com/journals/jp/2020/3631808/

Please revise the manuscript to rephrase the duplicated text, cite your sources, and provide details as to how the current manuscript advances on previous work. Please note that further consideration is dependent on the submission of a manuscript that addresses these concerns about the overlap in text with published work. We will carefully review your manuscript upon resubmission, so please ensure that your revision is thorough.

Reviewers' comments:

Reviewer's Responses to Questions

**Comments to the Author**

1. Is the manuscript technically sound, and do the data support the conclusions?

Reviewer #1: Yes

Reviewer #2: Yes

2. Has the statistical analysis been performed appropriately and rigorously? 

Reviewer #1: Yes

Reviewer #2: Yes

3. Have the authors made all data underlying the findings in their manuscript fully available?

Reviewer #1: Yes

Reviewer #2: Yes

4. Is the manuscript presented in an intelligible fashion and written in standard English?

Reviewer #1: Yes

Reviewer #2: Yes

5. Review Comments to the Author

Reviewer #1: This paper examined Partograph utilization and associated factors among obstetric care givers in

governmental health institutions of Jigjiga and Degehabur towns, Somali region,

Ethiopia. The background to the study was elaborated and gave specific reasons why Partograph utilization is important

I also found the approach and conclusions to be robust and useful. I do have little comments and suggestions.

Comments are attached in the manuscript

Reviewer #2: Dear Author,

I think your work will contribute to the literature. However, some corrections are needed. You can find my recommendations below.

I wish you good work…

1. Instead of “We aimed” in the abstract, “the purpose of study is…” should be written in a more academic language.

2. “In 2015 an estimated 303,000 women died because of pregnancy and childbirth-related complications.” If there is a more up-to-date statistic, it would be more appropriate to use it.

3. The problem description is well written.

4. “Study design and” and should be removed

5. study population

… Health professionals (Gynecologist and Obstetrician, students) and who did not attend labor cases were excluded from … this is not an exclusion criterion, it can be removed.

6. It is necessary to explain the sampling method with a more understandable and clear expression. In addition, it should be clearly stated how many people were included in the study in the Study Population.

7. “Semi - structured questionnaire which was developed from different literatures was used.” The sources used should be stated at the end of the sentence.

8. Ethics committee decision number should be attached.

9. Table 1 is in the wrong place in the text.

10. Sociodemographic findings should be Table 1 and the findings should be cited in parentheses accordingly. Table 2 should not be started.

11. Results- Socio-demographic characteristics of study participants –

Out of 235 study participants, 228… This section should be specified in the number of samples in the method.

12. In the findings, the sentence should not be started as “From 228 participants…”, a more academic language should be used.

13. In the findings, it is sufficient to write only the percentages instead of the frequencies. There should be no need to use parentheses.

14. In table 1 “Using partograph misleads management as the progress of labor and the partograph alert line are not aligned in most pregnant woman” percent sum is not 100. It should be corrected.

15. In the discussion section, first the findings of the study and then the findings in the literature should be included. Commentary sentences should be written after the studies are compared. The spelling of some sections is appropriate, but some require editing.

16. Suggestions should also be included in line with the results of the study. Missing suggestions should be added.

17. In Table 2: “List one complication” percent sum is not 100.

18. In the table titles, “N” should be written as “n” in lower case.

6. PLOS authors have the option to publish the peer review history of their article (what does this mean?). If published, this will include your full peer review and any attached files.

Reviewer #1: **Yes: **Dr. Angelina A. Joho

Reviewer #2: No

---

## [Author Response · Author response to Decision Letter 0]

23 Sep 2021

Comments raised by both reviewers and editor are important to enrich this research article. Authors have also looked each comments and modified the document accordingly.

---

## [Decision Letter · Decision Letter 1]

8 Nov 2021

PONE-D-21-20057R1Partograph utilization and associated factors among obstetric care givers in governmental health institutions of Jigjiga and Degehabur towns, Somali region, Ethiopia: a cross-sectional studyPLOS ONE

Dear Dr. Ayehubizu,

Thank you for submitting your manuscript to PLOS ONE. After careful consideration, we feel that it has merit but does not fully meet PLOS ONE’s publication criteria as it currently stands. Therefore, we invite you to submit a revised version of the manuscript that addresses the points raised during the review process.

We look forward to receiving your revised manuscript.

Kind regards,

Nülüfer Erbil, Ph.D, Prof.

Academic Editor

PLOS ONE

Journal Requirements:

Reviewers' comments:

Reviewer's Responses to Questions

**Comments to the Author**

1. If the authors have adequately addressed your comments raised in a previous round of review and you feel that this manuscript is now acceptable for publication, you may indicate that here to bypass the “Comments to the Author” section, enter your conflict of interest statement in the “Confidential to Editor” section, and submit your "Accept" recommendation.

Reviewer #1: All comments have been addressed

Reviewer #2: (No Response)

2. Is the manuscript technically sound, and do the data support the conclusions?

Reviewer #1: Yes

Reviewer #2: Yes

3. Has the statistical analysis been performed appropriately and rigorously? 

Reviewer #1: Yes

Reviewer #2: Yes

4. Have the authors made all data underlying the findings in their manuscript fully available?

Reviewer #1: Yes

Reviewer #2: Yes

5. Is the manuscript presented in an intelligible fashion and written in standard English?

Reviewer #1: Yes

Reviewer #2: Yes

6. Review Comments to the Author

Reviewer #1: I congratulate the author for responding to the comments given. Small observation, author should put space between words and also she/he should put all decimal points into two, this should be done in the whole document.

Thanks

Reviewer #2: Dear Author, you can find some sugesstions about article below:

• In the result section: In table 1 “Using partograph misleads management as the progress of labor and the partograph alert line are not aligned in most pregnant woman” percent sum is not 100. It should be corrected.

• There are two tables called Table 1.

• Suggestion statements should be added in the recommendation section.

• In Table 2: “List one complication” percent sum is not 100.

• Table 3 should be written, instead of table 4.

• In the table titles, “N” should be written as “n” in lower case.

7. PLOS authors have the option to publish the peer review history of their article (what does this mean?). If published, this will include your full peer review and any attached files.

Reviewer #1: **Yes: **Dr. Angelina A Joho

Reviewer #2: No

---

## [Author Response · Author response to Decision Letter 1]

23 Dec 2021

Response to Reviewer 1 comments:

Author should put space between words and also she/he should put all decimal points into two, this should be done in the whole document.

Response: I accepted and modified the document according to your comments.

Response to Reviewer 2 comments:

 In the result section: In table 1 “Using partograph misleads management as the progress of labor and the partograph alert line are not aligned in most pregnant woman” percent sum is not 100. It should be corrected.

Response: It is found in table 3 of the manuscript and the percent sum is 100 %( 7+9.7+21.9+26.3+35.1=100) 

 There are two tables called Table 1.

Response: It was corrected.

Suggestion statements should be added in the recommendation section.

Response: We took some modification

 In Table 2: “List one complication” percent sum is not 100. 

Response: Corrected based on your comment accordingly

91.8+8.2=100

Table 3 should be written, instead of table 4.

Response: We put table 3 and table 4 based on their logical order.

 In the table titles, “N” should be written as “n” in lower case

Response: It was corrected based on previous comment

---

## [Decision Letter · Decision Letter 2]

10 Feb 2022

Partograph utilization and associated factors among obstetric care givers in governmental health institutions of Jigjiga and Degehabur towns, Somali region, Ethiopia: a cross-sectional study

PONE-D-21-20057R2

Dear Dr. Ayehubizu,

We’re pleased to inform you that your manuscript has been judged scientifically suitable for publication and will be formally accepted for publication once it meets all outstanding technical requirements.

Kind regards,

Nülüfer Erbil, Ph.D, Prof.

Academic Editor

PLOS ONE

---

## [Editor Report · Acceptance letter]

24 Feb 2022

PONE-D-21-20057R2 

Partograph utilization and associated factors among obstetric care givers in governmental health institutions of Jigjiga and Degehabur towns, Somali region, Ethiopia: a cross-sectional study 

Dear Dr. Ayehubizu:

I'm pleased to inform you that your manuscript has been deemed suitable for publication in PLOS ONE. Congratulations! Your manuscript is now with our production department. 

Kind regards, 

on behalf of

Dr. Nülüfer Erbil 

Academic Editor

PLOS ONE